# Adapting Generalist Robot Policies with Semantic Reinforcement Learning

Jagdeep Singh Bhatia, Andrew Wagenmaker, William Chen, and Sergey Levine
U.C. Berkeley
semantic-action-rl.github.io

*Abstract*—**Generalist robot policies learn a diverse repertoire of behaviors from large-scale pretraining. In principle, this makes them excellent priors for downstream adaptation via reinforcement learning (RL). In practice, however, standard RL methods leveraging this prior optimize directly over robot actions, requiring the base policy's action distribution to be close to that of a performant policy from the start. This assumption breaks down for complex or long-horizon tasks, where policy inputs fall completely outside the pretraining distribution. Our key insight is that, for sufficiently expressive generalist policies, language prompts are an effective, alternative space for learning to solve such tasks: modulating language inputs elicits skills already within the policy's repertoire which can be composed to solve tasks beyond its zero-shot capabilities. We propose Semantic Action Reinforcement Learning (SARL), which learns to optimize this prompt space through online interaction, treating the generalist policy as a controllable skill prior. Importantly, leveraging pretrained skills rather than learning new ones from scratch yields structured, semantically meaningful exploration and highly efficient online improvement. In addition, learning to modulate prompts through experience grounds them in induced real-world behaviors for robust task-solving. Across real-world settings and simulated benchmarks, SARL unlocks fundamentally new capabilities—generalizing to complex, long-horizon tasks—and significantly outperforms existing approaches for improving robot behavior in deployment.**

## I. INTRODUCTION

Vision-language-action (VLA) models can learn broad skill repertoires from pretraining [5, 9, 38, 63], providing powerful priors over plausible robot behaviors. However, leveraging VLAs to solve new tasks remains challenging. While we could directly prompt models to carry out tasks of interest, this strategy requires models to not only have the necessary skills in their repertoire, but to also deploy them correctly. Particularly when we are interested in complex and long-horizon prompts, this presents a challenge — not only in the semantics of breaking down long tasks into stages, but also in *grounding* tasks in skills the robot can perform successfully. How can we better leverage the prior knowledge in general-purpose robot policies to learn new and more complex skills effectively?

Our key insight is that, by learning to modulate a VLA's prompts, and grounding the behaviors these prompts induce through real-world interaction, we can achieve constrained, high-quality exploration that enables VLAs to solve new tasks through reinforcement learning (RL). In particular, modulating language inputs elicits skills already within a VLA's repertoire which can be composed to solve tasks beyond its zero-shot

capabilities. In addition, learning to modulate prompts through experience grounds each prompt in the real-world behaviors it induces, leading to robust task solving. Unlike prior RL for robotics approaches that typically operate at the *robot action* level, we propose learning at the *semantic* level, allowing us to effectively take advantage of behavioral priors encoded in VLAs. For example, to train an omelette-making robot, rather than learning precise low-level robot actions, we can simply learn semantic language commands to steer the VLA: "turn on the stove", "open the fridge", "grab the egg", and so on. Reusing pre-trained skills rather than discovering new ones from scratch presents a potentially much easier learning problem, since the search space is naturally constrained to task-appropriate skills, but still allows for improvement over the base policy by sequencing learned skills to adapt to the task at hand. For VLAs with sufficiently broad language-inducible skill sets, language-based steering produces exactly the kind of exploration RL demands: highly expressive yet always grounded in reasonable actions, enabling efficient learning.

Our proposed approach, Semantic Action Reinforcement Learning (SARL), instantiates this capability through an online RL loop that treats the VLA's language instructions as an action space. It learns, through interaction with the real world, which language prompts will actually lead the VLA to complete the task of interest. By treating the VLA's language prompt as an "action", we can lift the learning problem from low-level robot action space to semantic language space and, by grounding prompts in real-world experience, efficiently refine which of them actually lead to success.

The core contribution of our paper is adapting VLAs over their language inputs through RL, and demonstrating that such an approach can solve new tasks efficiently on a real-world robot deployment. Our results—both in simulation and on a real robot—highlight that SARL unlocks fundamentally new capabilities, enabling us to efficiently learn to solve complex, long-horizon tasks typically infeasible for existing RL approaches or for zero-shot VLA deployment. Finally, we investigate why SARL learns effectively, and find that it is not only because SARL decomposes complex goals into achievable skills, but that SARL also learns a grounded mapping between semantic actions and the physical behaviors they induce.

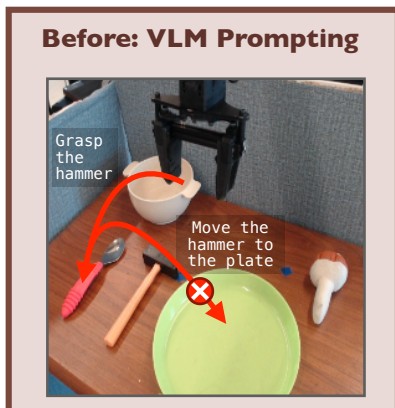
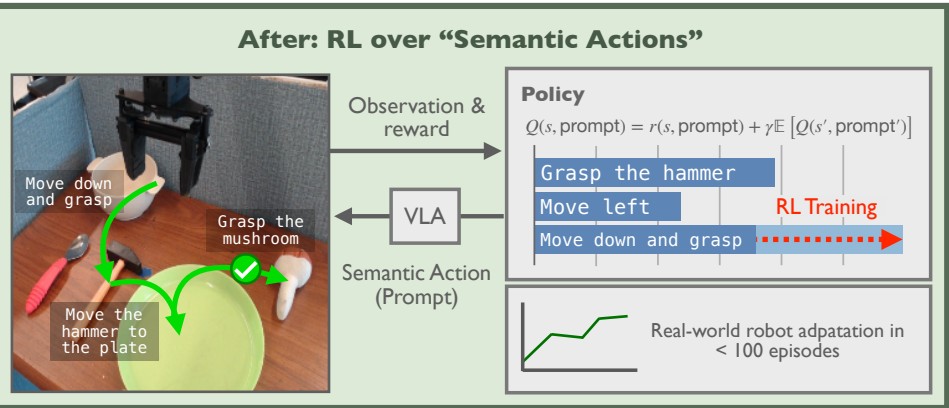

**Task Goal:** put the hammer on the plate and pick up the mushroom

Fig. 1: Steering VLA priors at deployment to solve complex, long-horizon tasks is challenging. To overcome this, we propose optimizing VLA language inputs via RL. The resulting *semantic* action space probes skills already encoded in VLAs from pretraining, yielding expressive exploration and efficient adaptation. However, not all induced behaviors make task progress: by learning to decompose complex tasks into *grounded* skills, our method enables real-robot adaptation in under 100 episodes.

## II. RELATED WORK

**RL improvement of VLAs.** The last several years have witnessed significant progress developing general-purpose language-conditioned robot policies such as vision-language-action models (VLAs) [11, 22, 75, 38, 9, 8, 74, 47, 63, 84]. While enabling effective task-solving in settings similar to their training data, VLAs often struggle to adapt to new tasks. To address this, a variety of approaches for RL improvement of VLAs have been proposed. These include training a residual correction policy [34, 4, 82, 35, 17, 80, 81, 71], steering the VLA's denoising process (when the VLA is a flow or diffusion model) [76], augmenting the VLA with "advantage conditioning" [3], and others [50, 56, 14, 28, 24, 48, 46]. However, these works all run RL at the level of *robot actions*, typically fixing the VLA's task prompt. In contrast, we run RL over the *VLA prompt itself* which, as we will see, unlocks novel task-solving abilities, enabling VLAs to generalize beyond single-task proficiency to solve novel long-horizon tasks. Our work is also somewhat related to hierarchical RL and skill learning [15, 40, 61, 65, 55, 20, 51, 2, 70, 62, 58, 79], where a high-level policy learns to direct a low-level policy to maximize reward. While conceptually similar, unlike these works we focus on the setting where the "low-level policy" is a VLA and the interface between both policies is language.

**Steering robots with language.** A variety of works consider how a language-conditioned policy's prompt should be chosen for successful robot task-solving. Early works in this vein explore using LLMs or VLMs to generate useful instructions for solving real-world tasks [29, 83, 30, 18], or rely on pre-programmed language-conditioned action primitives or "skill" policies as the language-to-robot interface [31, 44, 43, 85, 25]. More recently, several works have considered combining a VLM or LLM with a VLA or other language-conditioned policy, using the former to select the language prompts for the latter [1, 67, 68, 13]. Perhaps most similar to our work is [66]—which selects a VLA's language

command with a VLM, and utilizes the in-context learning ability of the VLM to adapt and improve the prompt based on past observations—and the work of [41], which trains a prompt-action verifier using offline data, and then select the prompt to use online from a fixed set of initial prompts based on verifier score. However, none of these works incorporate RL or other forms of online improvement which, as we will see, is critical to achieve robust task-solving behavior.

**Language-based RL outside of robotics.** While our focus is on robotic domains, a variety of other domains also seek to integrate RL and language. Our approach can be seen as a form of *prompt optimization*, commonly used to elicit desired capabilities from language models [69, 87, 64]. While a variety of prompt optimization approaches exist, most related to this work are RL-based methods [16, 86, 36, 78, 23, 49, 32, 7, 39, 42]. Outside of language models, language has also been used as a key tool in deep RL, enabling effective exploration [52, 54, 72, 19], reward shaping [12, 21], state abstraction design [60], and more [10, 27, 53, 57]. Though our work bears similarity to these works in that we also seek to optimize language prompts with RL and use language as an abstraction to enable efficient learning, the specific challenges we face—in particular, the need for high sample-efficiency in real-world robotic deployments—necessitates a very different algorithmic strategy than the ones proposed here.

## III. PRELIMINARIES

**Markov decision processes.** We consider decision-making in Markov decision processes (MDPs). MDPs are denoted by a tuple $\mathcal{M} = (\mathcal{S}, \mathcal{A}, P, P_0, \gamma, r)$, where $\mathcal{S}$ is the set of states, $\mathcal{A}$ the set of actions, $P : \mathcal{S} \times \mathcal{A} \to \triangle_{\mathcal{S}}$ the transition function, $P_0 \in \triangle_{\mathcal{S}}$ the initial state distribution, $\gamma \in [0, 1]$ the discount factor, and $r : \mathcal{S} \to \mathbb{R}$ the reward function. An episode consists of a sequence of interactions between an agent and the environment. Each episode starts at initial state $s_0 \sim P_0$, the agent selects action $a_0$, receives reward

$r(s_0)$, and transitions to state $s_1 \sim P(s_0, a_0)$, repeating this process until a terminal state is reached. A policy $\pi : \mathcal{S} \to \triangle_\mathcal{A}$ is a mapping from states to actions. The goal of RL is to learn a policy that maximizes the expected discounted reward, $V(\pi) := \mathbb{E}^\pi[\sum_t \gamma^t r(s_t)]$, where the expectation is over trajectories from rolling out $\pi$ in $\mathcal{M}$.

While usually taken as given, the parameterization of the action space $\mathcal{A}$ is often itself a key design decision in RL. In robotic control, $\mathcal{A}$ is most commonly taken to be the set of continuous robot actions (for example, joint positions, motor commands, or end-effector deltas), which we generically denote as $\mathcal{A}_{\mathrm{robot}}$. As we will discuss in the following, however, correctly choosing $\mathcal{A}$ and learning in the corresponding MDP can often enable much more efficient learning.

**Vision-language-action models (VLAs).** The robotics community has devoted significant attention to collecting diverse large-scale datasets, $\mathfrak{D} = \{(\tau_t, \ell_t)\}_t$, of human tele-operation trajectories, $\tau_t = (s_0^t, a_0^t, s_1^t, a_1^t, \ldots,)$ for $s_h^t \in \mathcal{S}$ and $a_h^t \in \mathcal{A}_{\mathrm{robot}}$, paired with corresponding language commands, $\ell_t$ [37, 77, 59]. Such datasets enable training language-conditioned robot policies—usually via behavioral cloning, that is, training a policy to produce action $a_h^t$ conditioned on state $s_h^t$ and language $\ell_t$. The resulting policies, given some task of interest described by a language command $\ell$, are able to produce actions $a_h \sim \pi_{\mathrm{vla}}(\cdot \mid s_h, \ell)$ that lead the robot to complete the task. Such "generalist" policies—more commonly referred to as "vision-language-action" models (VLAs)—have shown significant promise capturing language-controllable priors on effective robot behaviors, enabling general-purpose robotic task-solving abilities [5, 9, 38, 63].

## IV. LEVERAGING VLAS AS SEMANTIC ACTION PRIORS FOR EFFICIENT RL

In this section, we outline our approach, SARL, which leverages VLAs as controllable action priors that can be steered to enable effective online learning of complex tasks. In particular, we assume access to a VLA $\pi_{\mathrm{vla}}$ and are interested in solving some task $\tau$ corresponding to a reward function $r$. As illustrated in Figure 1, SARL treats the language prompt of a VLA as a "semantic action", and, through real-world interaction, learns to adaptively control this semantic action to steer the VLA to reliably complete tasks of interest.

### A. VLAs as Semantically Controllable Action Priors

In a typical VLA deployment, the user provides a fixed language command $\ell$ describing the task of interest $\tau$, and then uses a VLA to execute this command—in particular, for each observation $s$, the VLA generates action $a \sim \pi_{\mathrm{vla}}(\cdot \mid s, \ell)$, which is executed on the robot. While effective in settings where the observation and language command are supported by the VLA's training data $\mathfrak{D}$, this approach fails to enable VLAs to solve new tasks not covered in the task repertoire spanned by $\mathfrak{D}$.

In this work, we take an alternative view on VLA deployment. While a VLA may fail to execute a complex or out-of-distribution (OOD) command zero-shot, it can often produce reliable behaviors if prompted with simple, physically grounded instructions [13]. For instance, as illustrated in Figure 3, the VLA fails to execute both "put the hammer on the plate and pick up the mushroom" and "grasp the hammer". The first command is too complex, and the second lacks grounding — the hammer (likely an OOD object) is confused for a spoon. By contrast, "move down and grasp" is simple and avoids referring to the hammer by name, successfully steering the robot. While prompting the VLA with the former instructions will fail to result in successful task completion, adaptively controlling the VLA by selecting a sequence of low-level, grounded instructions *can* steer the robot to solve the overall task $\tau$. In other words, rather than viewing VLAs simply as policies to be statically prompted, we view them as *semantically controllable action priors* that can be dynamically guided throughout deployment to accomplish complex tasks of interest.

**Semantic MDPs.** This motivates a natural transformation of our original decision-making setting: rather than learning over the robot action space $\mathcal{A}_{\mathrm{robot}}$ as is typical in RL for robotics, we can learn in a *semantic* action space—the space of language commands, which we denote as $\mathcal{A}_{\mathrm{sem}}$—and deploy the VLA as a transformation between the two. Formally, rather than directly specifying action $a \in \mathcal{A}_{\mathrm{robot}}$, we can specify a "semantic action" $\ell \in \mathcal{A}_{\mathrm{sem}}$, generate robot action $a_{\mathrm{vla}} \sim \pi_{\mathrm{vla}}(\cdot \mid s, \ell)$ with our VLA, and transition to $s' \sim P(\cdot \mid s, a_{\mathrm{vla}})$. We denote this induced transition function as $P_{\mathrm{sem}} : \mathcal{S} \times \mathcal{A}_{\mathrm{sem}} \to \triangle_\mathcal{S}$, and the corresponding induced "semantic MDP" as $\mathcal{M}_{\mathrm{sem}} := (\mathcal{S}, \mathcal{A}_{\mathrm{sem}}, P_{\mathrm{sem}}, P_0, \gamma, r)$.

By considering decision-making in $\mathcal{M}_{\mathrm{sem}}$ instead of $\mathcal{M}$, we shift the burden from directly specifying precise robot actions to guiding the robot with semantic language commands, leveraging the VLA's priors to translate between semantic commands and robot actions. As described above, by correctly modulating these semantic actions, we can potentially steer the VLA to solve tasks that initially may appear outside its capabilities. Critically, learning over this semantic space enables efficient *semantic exploration*—instead of exploring the entirety of $\mathcal{A}_{\mathrm{robot}}$, we can focus our exploration only on the "semantically meaningful" behaviors encoded in the VLA's priors and accessed via language prompting. As we will see, this enables highly efficient exploration and online improvement.

### B. SARL: Learning in Semantic Action Spaces

Our proposed approach, SARL, aims to learn how to correctly select these "semantic actions" in order to steer VLAs to task success. While a VLA may act as an effective prior over robot actions, in general we do not know which language commands will effectively steer it to solve tasks of interest, and we must learn this through interaction with both the VLA and the environment. While in principle this can be achieved by applying any RL algorithm to learn in the semantic MDP $\mathcal{M}_{\mathrm{sem}}$, we propose a particular instantiation that seeks to enable efficient real-world learning.

## Base Policy
Zero-shot prompting fails on complex tasks.

## Semantic Action RL
Composing grounded skills is successful.

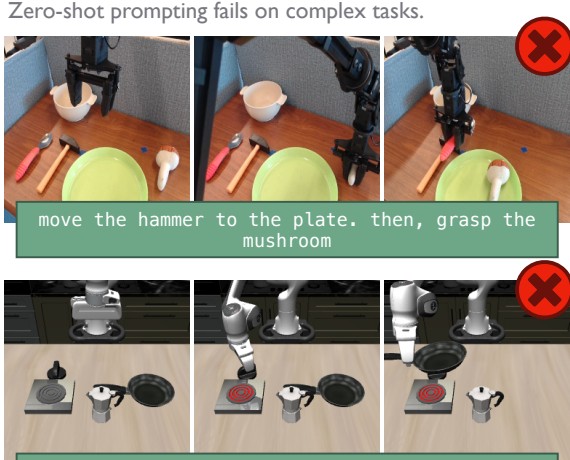
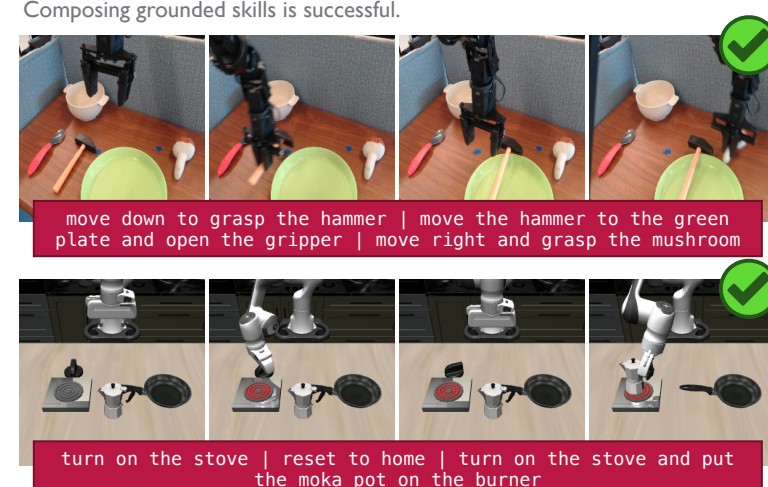

move the hammer to the plate. then, grasp the mushroom

move down to grasp the hammer | move the hammer to the green plate and open the gripper | move right and grasp the mushroom

turn on the stove and put the moka pot on it

turn on the stove | reset to home | turn on the stove and put the moka pot on the burner

Fig. 2: On complex, long-horizon tasks, the base policy fails when zero-shot prompted. SARL solves them through learned prompting, sequencing skills covered under the pretraining distribution.

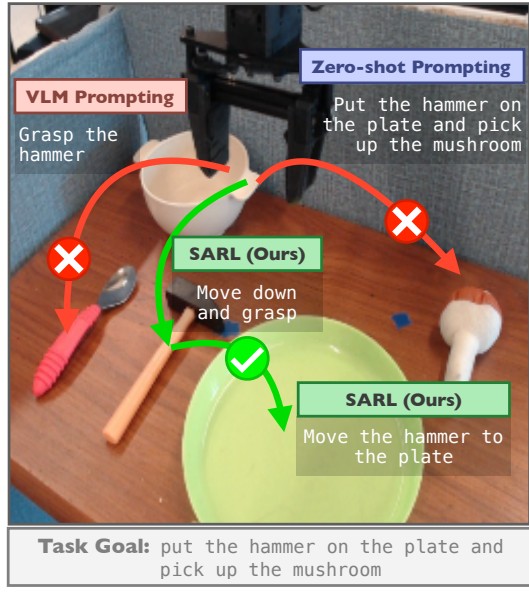

VLM Prompting
Grasp the hammer

Zero-shot Prompting
Put the hammer on the plate and pick up the mushroom

SARL (Ours)
Move down and grasp

SARL (Ours)
Move the hammer to the plate

Task Goal: put the hammer on the plate and pick up the mushroom

Fig. 3: Adapting VLAs to solve new tasks requires *decomposing* task-goals into achievable stages, and *grounding* each stage in behaviors that can be executed successfully in the environment. VLAs prompted zero-shot struggle with both decomposition and grounding, and prompting VLAs with VLMs achieves decomposition but not grounding. Only SARL achieves both by learning to optimize a VLA's prompt inputs with RL.

**Semantic Action Reinforcement Learning (SARL).** At a high level, SARL operates by selecting a semantic action $\ell_t \in \mathcal{A}_{\text{sem}}$ at each step $t$ and state $s_t$, taking semantic action $\ell_t$ in $\mathcal{M}_{\text{sem}}$ (or, equivalently, generating action $a_t \sim \pi_{\text{vla}}(\cdot \mid s_t, \ell_t)$ and taking action $a_t$ in $\mathcal{M}$), and observing reward $r_t$ and next state $s_{t+1}$. In order to determine which semantic action to take at each step, SARL learns a *semantic Q-function*, $Q_{\text{sem}}$, trained to estimate the effectiveness of semantic actions

at making progress towards completing $\tau$ from each state $s$ through temporal-difference (TD) backups [6]. When acting from state $s_t$, SARL samples a semantic action from the soft-max distribution induced by their $Q$-values, $\ell_t \sim \pi_{\text{sem}}^t(\cdot \mid s)$ for $\pi_{\text{sem}}^t(\ell \mid s) \propto \exp(Q_{\text{sem}}^t(s_t, \ell))$. A formal description of our approach can be found in Algorithm 1.

---

**Algorithm 1** Semantic Action Reinforcement Learning (SARL)

---

1: **input**: semantic environment $\mathcal{M}_{\text{sem}}$, semantic action set $\mathcal{A}_{\text{sem}}$
2: Initialize $Q_{\text{sem}}^1$ randomly, replay buffer $\mathfrak{B} \leftarrow \emptyset$
3: **for** $t = 1, 2, 3, \ldots$ **do**
4:     Set $\pi_{\text{sem}}^t(\ell \mid s) \propto \exp(Q_{\text{sem}}^t(s, \ell))$
5:     Sample $\ell_t \sim \pi_{\text{sem}}^t(\cdot \mid s_t)$
    // Equivalently, execute action $a_t \sim$
$\pi_{\text{vla}}(\cdot \mid s_t, \ell_t)$ in $\mathcal{M}$
6:     Execute semantic action $\ell_t$ in $\mathcal{M}_{\text{sem}}$, observe reward $r_t$ and next state $s_{t+1}$
7:     $\mathfrak{B} \leftarrow \mathfrak{B} \cup \{(s_t, \ell_t, r_t, s_{t+1})\}$
8:     Set $\bar{V}_{\text{sem}}^t(s') \leftarrow \mathbb{E}_{\ell' \sim \pi_{\text{sem}}^t(\cdot \mid s')}[\bar{Q}_{\text{sem}}^t(s', \ell')]$ and

$$Q_{\text{sem}}^{t+1} \leftarrow \min_Q \sum_{(s, \ell, r, s') \in \mathfrak{B}} (Q(s, \ell) - r - \bar{V}_{\text{sem}}^t(s'))^2$$

9: **end for**

---

**Compressing $\mathcal{A}_{\text{sem}}$ with VLMs.** Naïvely learning over the space of all possible VLA prompts ($\mathcal{A}_{\text{sem}}$) is intractable. To mitigate this, SARL refines $\mathcal{A}_{\text{sem}}$ and reduces the search space by relying on Internet-pretrained models, which often encode powerful semantic priors. In particular, vision-language models (VLMs) have proved effective at proposing semantic commands relevant for robotic control [13, 66]. Instead of searching over all possible prompts, SARL queries a VLM at current state $s_t$ with high-level task $\tau$ to generate a

set of *candidate* semantic actions, $\mathcal{A}^t_{\mathrm{sem}}$. Critically, $\mathcal{A}^t_{\mathrm{sem}}$ is used instead of $\mathcal{A}_{\mathrm{sem}}$ in Algorithm 1, enabling SARL to be computationally tractable.

Note that, while VLMs can often generate reasonable candidate VLA prompts, they fundamentally lack grounding for which commands are actually effective. A potential command might appear reasonable to a VLM, yet the VLA's exact response to this command is highly dependent on the deployment setting, the VLA's training data, and other factors the VLM does not have access to. In other words, VLMs alone are not effective at directly controlling VLAs (as we will see in Section V). By interleaving VLM queries with real-world interaction, SARL is able to achieve the best-of-both worlds—effectively leveraging the VLM's semantic priors, while also enabling improvement over these priors by grounding semantic actions in physical VLA behaviors from experience.

## V. EXPERIMENTS

Our experiments evaluate SARL both as a sample-efficient algorithm for adapting VLA priors and as a method that steers VLAs through high-level language instructions. We aim to answer three questions. First, can SARL's semantic exploration space efficiently adapt VLAs to solve new tasks? Second, compared to traditional action-space steering methods (e.g., residual RL [80, 4, 81] or latent-noise space steering [76]), does operating in a lifted, semantic prompt space allow SARL to efficiently solve complex, long-horizon tasks where these prior methods may struggle? Finally, does optimizing prompts via online RL outperform non-RL VLA prompting methods (e.g., in-context learning (ICL) with VLMs), by explicitly grounding semantic instructions in the physical behaviors they induce? We investigate these questions across a suite of challenging multi-step tasks within both the Libero simulation benchmark [45] and a real-world WidowX robot deployment.

### A. Experimental Setup

We evaluate SARL on the simulated Libero-10 benchmark [45] and four challenging real-world tasks [77] shown in Figure 6. These tasks are specifically designed to be long-horizon, and involve multiple steps. The policies we use — based on $\pi_{0.5}$ [63], and finetuned on the Libero-90 and DROID datasets [45, 37], and the Bridge dataset [77], respectively — perform poorly on these tasks, only achieving significant performance on one task in each setting. Consequently, we posit that these tasks present a good evaluation domain for RL methods that leverage pre-trained generalist policies: while they contain many atomic behaviors that are well-represented in Libero and Bridge's training sets, they may reveal the difficulty of applying these behaviors directly to complex new tasks. This is exactly the challenge that our method aims to solve.

In addition to the original language prompt space for the VLA, in Libero we also introduce a "reset-to-home" command that resets the robot arm to its starting state. Note that this is straightforward to implement on a real robot, and the capacity

to seamlessly integrate such auxiliary controllers highlights a unique strength of our approach.

We compare SARL against two classes of baseline approaches. As VLA RL baselines, we consider Diffusion Steering via Reinforcement Learning (DSRL) [76], which learns to steer diffusion policies in their latent-noise space, and Residual RL [80, 4, 81], which learns a residual policy that predicts small corrective actions which are added to the base policy before execution. For both baselines we rely on standard implementations. Note that, to the best of our knowledge, all existing uses of DSRL and Residual RL deploy with a *fixed* VLA prompt; as such, we adopt the same strategy, using task prompts indicated in Figure 6 for real, and provided Libero task prompts in sim. For our second class of baselines, we explore approaches that adaptively adjust the VLA's prompt. In particular, we consider an in-context learning-based VLM approach that processes past observations in-context and uses this information to select prompts that steer the VLA to task completion [13, 66]. For SARL, we utilize the VLM-based restriction of $\mathcal{A}_{\mathrm{sem}}$ outlined in the preceding section. To simplify the prompt space further, we only consider the first $N \in \{32, 64, 100\}$ prompts proposed by the VLM; we find this to produce sufficiently expressive behaviors to enable effective learning.

Finally, a core advantage of language-based steering is the simplicity of integrating human demonstration: operators merely provide the VLA with periodic language instructions. We collect three full demonstrations per WidowX task and seed them into SARL and Residual RL's replay buffers prior to training. Because DSRL inherently cannot ingest such demonstrations, it is evaluated without them. Further experimental details are provided in Appendix A.

### B. SARL Enables Efficient Adaptation of Generalist Policies

We first evaluate SARL's ability to learn effective task-solving behaviors. As shown in Figures 4 and 5, SARL is able to significantly improve the performance of the base VLA across Libero-10 and four challenging long-horizon real-world tasks. In all settings, SARL is able to improve from an initial success rate of near-0% up to 80% after only 60-100 online episodes, suggesting that the VLA's prompts are an expressive and high-quality exploration space for task-learning. This is even true for tasks where the base policy is not able to sample a single success, suggesting effective adaptation. Notably, our approach improves over traditional action-space steering methods, such as DSRL and Residual RL, which are unable to learn at all on several tasks. SARL also significantly outperforms the ICL VLM baseline, which adaptively selects VLA prompts by in-context learning over a history of interactions. These results highlight SARL's effectiveness as a method for adaptation on real-world robots, enabling novel task-solving abilities in VLAs.

We emphasize that the tasks in our evaluation suite are unique in their complexity and horizon for deployment-time adaptation compared to prior works [76, 81, 80]. Our tasks, which by design require invoking sequences of skills seen

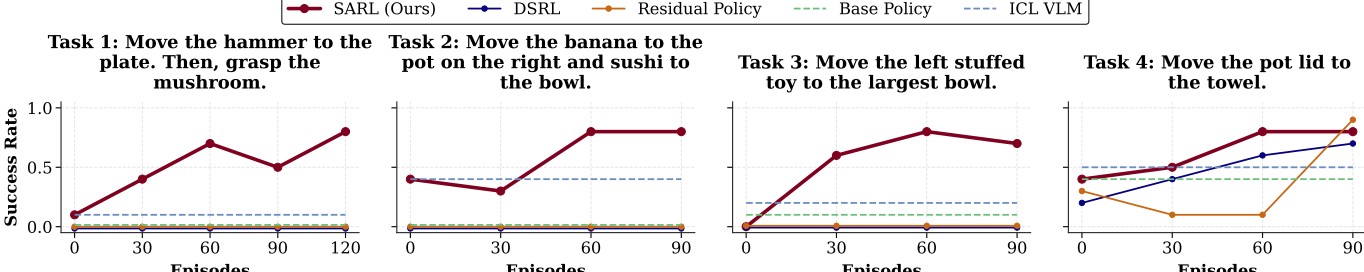

Fig. 4: Across four complex, real-world tasks, learning over semantic actions with SARL enables the best improvement of generalist policy behavior in deployment. SARL enables new capabilities over prior steering methods such as DSRL [76] and Residual RL [80, 4, 81], as these methods are fundamentally constrained by the base policy's performance under the task's original prompt. In particular, both DSRL and Residual RL perform best on Task 4, which has the least complex task goal. Finally, our method outperforms an in-context learning VLM baseline (ICL VLM) [13], since VLMs struggle to ground semantic actions in physical behaviors. Each data-point represents 10 evaluations.

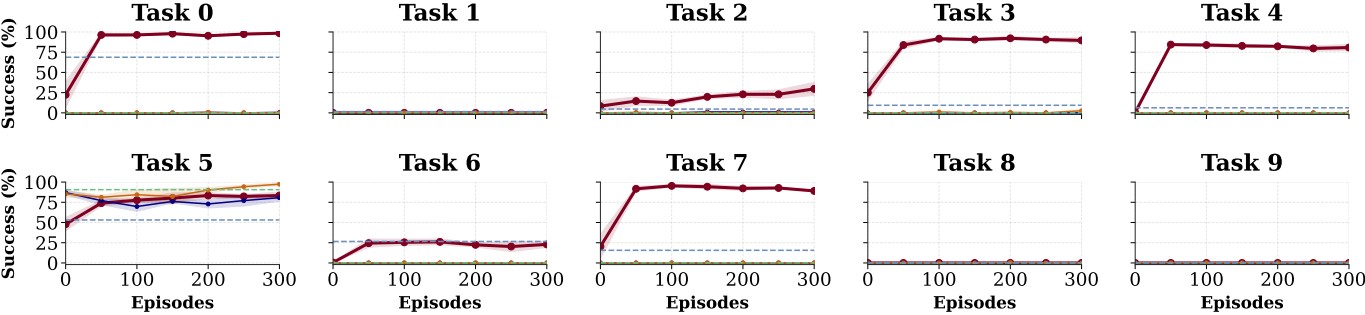

Fig. 5: SARL performs best among prior methods for deployment-time adaptation — DSRL [76], Residual RL [80, 4, 81], and an in-context learning VLM [13] — on average across out-of-distribution Libero-10 [45] long-horizon tasks. Our method successfully adapts a policy [33] on five tasks, and matches performance on another already close to solved. Four tasks remain unsolved by any method. Each graphed data-point represents 64 evaluations and standard error over 3 seeds.

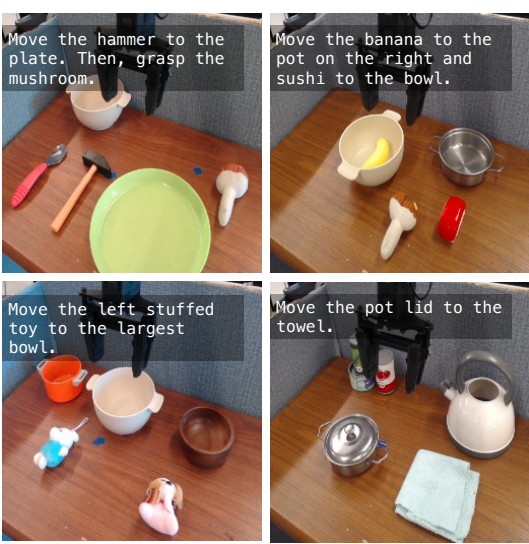

Fig. 6: We test our approach on four complex, long-horizon tasks along with the simulated Libero-10 (Libero-long) benchmark [45]. These tasks require composing skills seen during pretraining, making them a good evaluation domain for algorithms that aim to adapt pre-trained generalist policies.

during pretraining, are not able to be solved by action-space steering methods, such as DSRL and Residual RL. The uniquely ability of SARL to solve these tasks highlights that it unlocks a completely new capability over these prior steering methods.

### C. SARL Unlocks Fundamentally New Adaptation Ability

As shown in Figures 4 and 5, SARL significantly outperforms standard steering methods (DSRL and Residual RL) operating over robot actions directly. We hypothesize that this is because 1) existing steering methods are constrained to a single task prompt and 2) the VLA's action distribution under this task prompt does not adequately cover the diversity of actions needed to solve the task, making action-based steering ineffective.

Indeed, on the multi-step prompts of Libero-10 and our real-world task suite, the base policy's action distribution collapses into entirely incorrect modes, far from an optimal policy's distribution. A qualitative example is shown in Figure 2, where a VLA instructed to "move the hammer to the plate and grab the mushroom" (Row 1), instead moves the mushroom to the plate and grabs a spoon. This property is catastrophic for methods that steer over actions directly. For instance, DSRL can filter to find good actions from the base policy's distribution, but cannot synthesize fundamentally new ones.

## In-Context Learning VLM
Actions lack grounding.

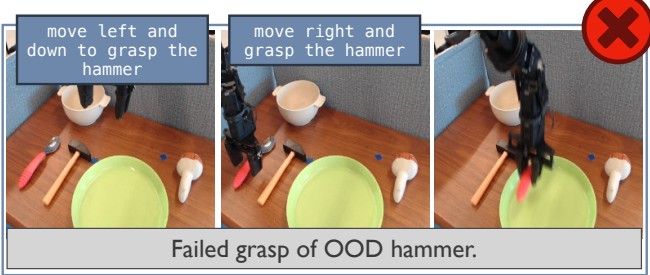

Failed grasp of OOD hammer.

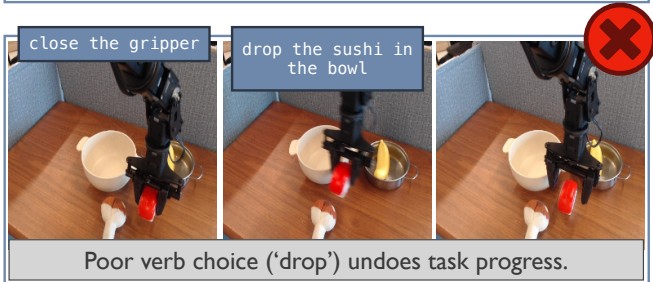

Poor verb choice ('drop') undoes task progress.

## Semantic Action RL
Learned, grounded actions.

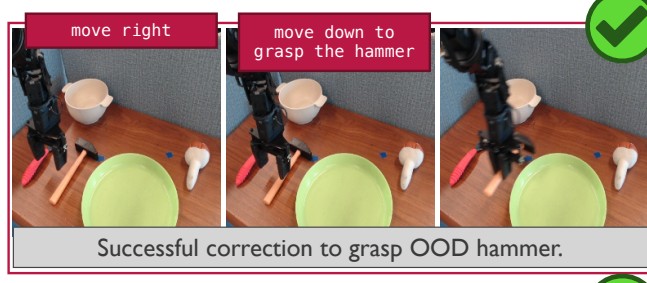

Successful correction to grasp OOD hammer.

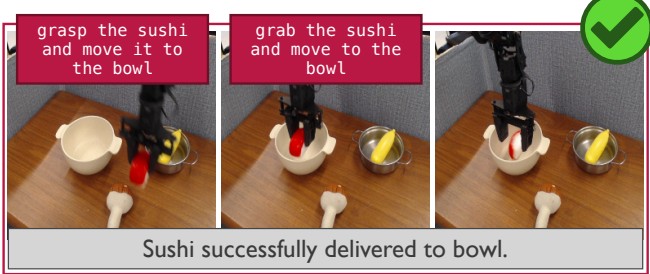

Sushi successfully delivered to bowl.

Fig. 7: Across tasks, an in-context learning VLM [13] takes semantic actions that are semantically meaningful, but lack grounding in physical behaviors, leading to task failures. By contrast, SARL's learned semantic actions correctly navigate scenarios where choosing the right instruction is critical. Additional examples can be found in Figure 9.

Similarly, residual RL is restricted to exploring a narrow funnel around the base policy's actions. Consequently, these baselines only learn effectively when the base policy already exhibits close to successful behaviors on the underlying task (e.g., Libero Task 5 or real-world Task 4), but cannot recover from the significantly incorrect behavior modes observed when task prompts are out-of-support. In contrast, SARL is not constrained in this way. By modulating prompts, SARL explores regions of the VLA's behavioral prior that remain entirely inaccessible to action-only steering methods. This capability is critical for solving long-horizon tasks, and SARL uniquely drives significant progress across five Libero-10 tasks, along with all tested WidowX tasks.

### D. Instruction Grounding is as Important as Decomposition

Our results show that SARL enables highly efficient learning of complex, long-horizon tasks even when task-prompts fall outside the base policy's support. To disentangle whether this capability stems from more than just instruction decomposition (ie. breaking down task prompts into sequences of short-horizon commands), we consider a closer comparison of our approach against its VLM baseline [13], which selects instructions via in-context learning (ICL) over interaction history.

As shown in Figures 4 and 5, the VLM baseline outperforms the zero-shot base policy, confirming that task decomposition does improve performance. However, across the board, SARL significantly outperforms the VLM. Qualitative examples in Figure 7 illustrate why: the VLM frequently selects commands that seem semantically plausible but still cause execution failures. For instance, in Row 1, the VLM is unable to pick the hammer (likely an OOD object). Referencing the "hammer"

by name mistakenly causes the VLA to grasp a spoon, and only spatial commands (e.g., "move right", "move down") successfully guide the VLA. In another instance (Row 2), the VLM selects an incorrect verb ("drop"), causing the VLA to prematurely release a piece of sushi before reaching the target. While the VLM can learn in-context, it cannot recover from such failures in a single episode. By contrast, through many episodes of experience, SARL is able to learn the grounded behavior induced by each language command, and uses this knowledge to select language commands appropriately. Our results highlight that this ability is critical for performant adaptation.

### VI. DISCUSSION AND LIMITATIONS

In this work we propose SARL, a new method for steering VLAs over their language inputs. Our results illustrate that SARL enables highly efficient learning, particularly excelling at complex, long-horizon tasks where standard steering methods are unsuccessful. We additionally disentangle that SARL's performance comes from both semantically decomposing task goals using a VLM, while also learning to ground those behaviors in the actions they induce through experience. We find SARL to be a powerful yet efficient approach for adapting generalist robot policies online, unlocking new capaiblities over prior methods. The two main limitations of SARL are speed (due to having a VLM in the loop), and the prerequisite of having VLAs that produce diverse behaviors when prompted with diverse instructions. For the former, an interesting extension of our work is studying whether VLMs are required at deployment time, or if the learned $Q$-function can generalize. For the later, we believe the community will continue to develop stronger language-conditioned policies,

making our work increasingly relevant, and hope that SARL will also inspire their training.

ACKNOWLEDGMENTS

Redacted to preserve anonymity for review.

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

### A. SARL: In-practice Optimizations

We make three optimizations to Algorithm 1 presented in Section IV-B to improve learnability and speed.

First, as discussed in Section IV-B, naïvely learning over the space of all possible VLA prompts ($\mathcal{A}_{\text{sem}}$) is intractable. Instead, at every execution step, we use a VLM to produce a set of candidate semantic actions $\mathcal{A}_{\text{sem}}^t$, which SARL is then run over. However, since the VLM can still, in principle, generate any language prompt, this approach requires a prompt featurization scheme that can both 1) encode the semantic meaning of each prompt 2) generalize the VLA's grounded behavior resulting from each prompt across these features. In practice, we find the latter to be especially difficult with off-the-shelf embeddings.

As an alternate approach, we cache the first $k$ prompts seen by the algorithm (either generated by the VLM, or provided by a human during demonstrations). After the cache has been filled, the VLM is queried to generate $\mathcal{A}_{\text{sem}}^t$, but may only choose commands from the size-$k$ cache, rather than generated them open-endedly. This allows for a one-hot prompt featurization which we find enables efficient learning. $|\mathcal{A}_{\text{sem}}^t|$ and $k$ for each task can be found in Table IV as `outer_policy.max_candidates` and `outer_policy.cache_size`, respectively.

We consider two additional optimizations: First, for each next state $s_{t+1}$ added to the replay buffer in Algorithm 1, we cache the corresponding generated $\mathcal{A}_{\text{sem}}^{t+1}$, which avoids the need to call the VLM during Temporal-Difference (TD) backups [69]. Second, instead of generating $\mathcal{A}_{\text{sem}}^t$ for each $t$, we use the same $\mathcal{A}_{\text{sem}}^t$ for $t_{\text{vlm}}$ steps before regenerating — this speeds up the time to collect each episode. A revised Algorithm 1 with these optimizations is provided in Algorithm 2.

### B. MDP Definition

For steering methods that make decisions over chunks of low-level actions, it is helpful to distinguish two distinct Markov Decision Processes (MDPs): low-level MDPs and high-level MDPs. In low-level MDPs, each transition corresponds to executing a single action in the underlying environment. By contrast, high-level MDPs operate over a coarser temporal resolution, where each transition corresponds to a "chunk" of low-level actions executed entirely open-loop. When converting low-level MDPs into high-level MDPs, we define a *high-level step size* to be the number of low-level actions executed open-loop in each transition of the high-level MDP. The horizon of a task in a high-level MDP is therefore its horizon in the corresponding low-level MDP divided by the high-level step-size.

The methods evaluated in this work operate over high-level MDPs, which improves rollout speed (especially when querying a VLM is required at each MDP step) and temporal consistency of actions. As shown in Table I, each task's horizon — defined in terms of low-level environment steps — remains constant across all compared methods. However, the high-level step size varies across tasks and methods.

For real-world WidowX experiments, we use a high-level step size of 20 for the VLM baseline, querying the VLM every 20 environment steps for a new prompt to issue the VLA. This keeps in line with Chen et al. [13], which this baseline is based on. For our method, SARL, we query the VLM at the same rate, every 20 steps ($t_{\text{vlm}} = 20$), to generate candidate semantic actions $\mathcal{A}_{\text{sem}}^t$. However, we use a high-level step size of 1, allowing the RL policy fine-grained control to choose which of these candidates is most relevant at each state. For a fair comparison, we also set the high-level step size of DSRL [76] and Residual RL baselines [80, 4, 81] to be 1, so that they are also able to make reactive, fine-grained adjustments based on the state.

In simulation, we sweep the VLM's high-level step size over 40, 80 and 120 and find 40 to be best. We additionally sweep SARL's high-level step size over 40 and 120 (keeping $t_{\text{vlm}} = 1$) and find better mode-consistent exploration with 120. The original DSRL implementation [76] uses $\pi_0$ [9] with a high-level step size of 20. Our work uses $\pi_{0.5}$ [63] — for improved language-following capabilities — with a high-level step size of 10, as this is the action horizon of $\pi_{0.5}$. For fairness, we use the same step size for Residual RL. We find our hyperparameter selection for both DSRL and Residual RL to be competitive with the Libero [45] results from Wagenmaker et al. [76]. Further details on hyperparamters for DSRL [76], Residual RL [80, 4, 81], and SARL can be found in Tables II, III, and IV, respectively.

### C. Reset-to-home

In simulation, we evaluate SARL along with baselines on the Libero-10 (Libero-long) benchmark [45]. As mentioned in Section V-A, we use a VLA based on $\pi_{0.5}$ [63], and finetuned on the Libero-90 and DROID datasets [45, 37]. In addition to the original language prompt space for this VLA, we also introduce a "reset-to-home" command that resets the robot arm to its starting state. We find such a command to be necessary to chain individual skills and make progress on the long-horizon tasks of the Libero-10 benchmark. Note that the "reset-to-home" functionality is straightforward to implement on both simulated and real robots, since it simply runs a PID controller to match the robot's joint-angles or end-effector pose at its starting configuration. Note that the capacity to seamlessly integrate such auxiliary controllers through language highlights a unique strength of our approach. Both SARL and the VLM baseline may leverage the "reset-to-home" command, and it is up to the individual approach how many steps to run this instruction for.

### D. Environment Step Plots

In the main paper, we provide plots which measure performance against the number of episodes spent training. In Figure 8, we provide the same plot with the x-axis swapped to show the number of environment steps spent training, as this is a standard choice in many prior works.

---

**Algorithm 2** Semantic Action Reinforcement Learning (SARL) with Optimizations

---

1: **input**: semantic environment $\mathcal{M}_{\text{sem}}$, semantic action set $\mathcal{A}_{\text{sem}}$, cache size limit $k$, regeneration period $t_{\text{vlm}}$, demonstrations $\mathfrak{D}$ (optional).
2: Initialize $Q_{\text{sem}}^1$ randomly, replay buffer $\mathfrak{B} \leftarrow \emptyset$, prompt cache $\mathcal{C} \leftarrow \emptyset$. If demonstrations $\mathfrak{D}$ are provided, update $\mathfrak{B}$ and $\mathcal{C}$ accordingly.

3: **function** GETCANDIDATES($s, t, \mathcal{C}, \mathcal{A}_{\text{sem}}^{\text{prev}}$)
4:     **if** $\texttt{mod}(t-1, t_{\text{vlm}}) == 0$ **then**
        `// Refresh candidate actions`
5:         **if** $|\mathcal{C}| < k$ **then**
            `// Cache not yet full: generate open-endedly and cache`
6:             Query VLM open-endedly for candidate prompts $\mathcal{A}_{\text{sem}}^{\text{new}}$ given $s$
7:             $\mathcal{C} \leftarrow \mathcal{C} \cup \mathcal{A}_{\text{sem}}^{\text{new}}$
8:         **else**
            `// Cache is full: restrict VLM choices to the cache`
9:             Query VLM to select candidate prompts $\mathcal{A}_{\text{sem}}^{\text{new}} \subseteq \mathcal{C}$ given $s$
10:         **end if**
11:         **return** $\mathcal{A}_{\text{sem}}^{\text{new}}$
12:     **else**
        `// Reuse previous step's candidates`
13:         **return** $\mathcal{A}_{\text{sem}}^{\text{prev}}$
14:     **end if**
15: **end function**

16: $\mathcal{A}_{\text{sem}}^1 \leftarrow$ GETCANDIDATES($s_1, 1, \mathcal{C}, \emptyset$)
17: **for** $t = 1, 2, 3, \dots$ **do**
18:     Set $\pi_{\text{sem}}^t(\ell \mid s_t; \mathcal{A}_{\text{sem}}^t) \propto \exp(Q_{\text{sem}}^t(s_t, \ell))$ for $\ell \in \mathcal{A}_{\text{sem}}^t$
19:     Sample $\ell_t \sim \pi_{\text{sem}}^t(\cdot \mid s_t; \mathcal{A}_{\text{sem}}^t)$
    `// Equivalently, execute action` $a_t \sim \pi_{\text{vla}}(\cdot \mid s_t, \ell_t)$ `in` $\mathcal{M}$
20:     Execute semantic action $\ell_t$ in $\mathcal{M}_{\text{sem}}$, observe reward $r_t$ and next state $s_{t+1}$
21:     $\mathcal{A}_{\text{sem}}^{t+1} \leftarrow$ GETCANDIDATES($s_{t+1}, t+1, \mathcal{C}, \mathcal{A}_{\text{sem}}^t$)
    `// Cache` $\mathcal{A}_{\text{sem}}^{t+1}$ `to eliminate VLM calls during TD backups`
22:     $\mathfrak{B} \leftarrow \mathfrak{B} \cup \{(s_t, \ell_t, r_t, s_{t+1}, \mathcal{A}_{\text{sem}}^{t+1})\}$
23:     Set $\bar{V}_{\text{sem}}^t(s') \leftarrow \mathbb{E}_{\ell' \sim \pi_{\text{sem}}^t(\cdot \mid s'; \mathcal{A}_{\text{sem}}')}[\bar{Q}_{\text{sem}}^t(s', \ell')]$ where $\pi_{\text{sem}}^t(\ell' \mid s'; \mathcal{A}_{\text{sem}}') \propto \exp(Q_{\text{sem}}^t(s', \ell'))$ for $\ell' \in \mathcal{A}_{\text{sem}}'$ and

$$Q_{\text{sem}}^{t+1} \leftarrow \min_Q \sum_{(s, \ell, r, s', \mathcal{A}_{\text{sem}}') \in \mathfrak{B}} (Q(s, \ell) - r - \bar{V}_{\text{sem}}^t(s'))^2$$

24: **end for**

---

TABLE I: **MDP Definition Across Tasks and Methods.** Task horizon is the total number of low-level environment steps allocated per task. The high-level step size defines the conversion between high-level steering steps to low-level environment steps. To get the horizon $H$ of the high-level MPD for each method and task (over which learning happens), divide the task horizon by the high-level step size.

| Task | Task Horizon | | | | High-Level Step Size | | | |
|---|---|---|---|---|---|---|---|---|
| | SARL | DSRL | Residual | VLM | SARL | DSRL | Residual | VLM |
| **Real Task 1:** Move the hammer to the plate. Then, grasp the mushroom. | 120 | 120 | 120 | 120 | 1 | 1 | 1 | 20 |
| **Real Task 2:** Move the banana to the pot on the right and sushi to the bowl. | 160 | 160 | 160 | 160 | 1 | 1 | 1 | 20 |
| **Real Task 3:** Move the left stuffed toy to the largest bowl. | 100 | 100 | 100 | 100 | 1 | 1 | 1 | 20 |
| **Real Task 4:** Move the pot lid to the towel. | 60 | 60 | 60 | 60 | 1 | 1 | 1 | 20 |
| **Libero-10** | 600 | 600 | 600 | 600 | 120 | 10 | 10 | 40 |

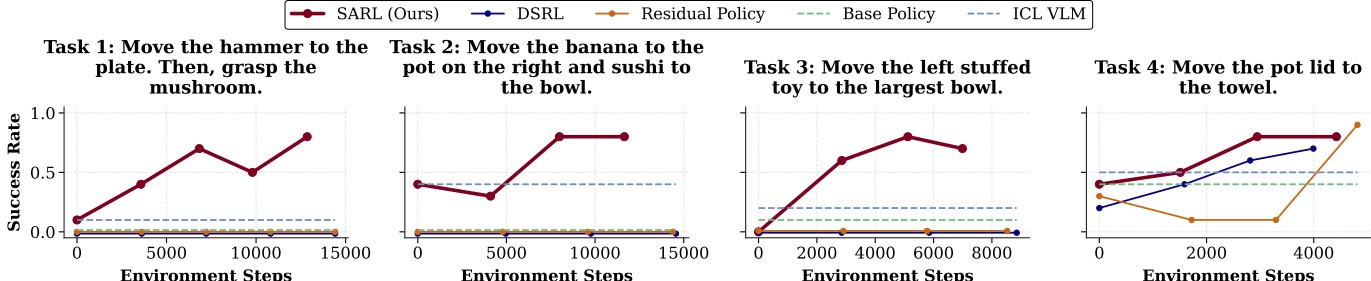

Fig. 8: Across four complex, real-world tasks, learning over semantic actions with SARL enables the best improvement of generalist policy behavior in deployment. SARL enables new capabilities over prior steering methods such as DSRL [76] and Residual RL [80, 4, 81], as these methods are fundamentally constrained by the base policy's performance under the task's original prompt. In particular, both DSRL and Residual RL perform best on Task 4, which has the least complex task goal. Finally, our method outperforms an in-context learning VLM baseline (ICL VLM) [13], since VLMs struggle to ground semantic actions in physical behaviors. Each data-point represents 10 evaluations. This plot mirrors Figure 4, but the x-axis is changed to measure environment steps instead of episodes.

### E. VLM Prompts

We use the Gemini model family [73, 74] for all VLM calls in SARL and the VLM baseline. All VLM calls use the `gemini-3-flash-preview` model and are allocated $1024$ thinking tokens. Our methods use six total prompts (across real and sim experiments, two prompts generate candidate semantic actions $\mathcal{A}^t_{\text{sem}}$ open-endedly and from a cache, and another is used for the in-context learning VLM baseline). All prompts make minimal modifications to Chen et al. [13]. Prompts for real are provided in Figures 10, 11, and 12 and those for sim are provided in Figures 13, 14, and 15.

### F. Additional Qualitative Examples

Additional qualitative examples of catastrophic failures due to the in-context learning VLM baseline's lack of grounding are shown in Figure 9 (extending Figure 7).

TABLE II: **DSRL Hyperparameters.** Implementations make minimal modifications to Wagenmaker et al. [76]. *Real* indicates experiments run on the WidowX while *Sim* indicates experiments run on Libero-10 [45].

| Hyperparameter | Value | Description |
|---|---|---|
| `outer_policy.action_dim` | 32 | Action space dimension, which is the same as the latent-noise dim for $\pi$ series policies [63]. This low-dim action is repeated across the predicted action horizon as done in [76] |
| `outer_policy.state_dim` | 8 | Policy has access to a proprioceptive observation |
| `outer_policy.policy_chunk_size` | *Real:* 1 │ *Sim:* 10 | Number of actions used from each evaluation of $\pi$ series policies [63] which generate chunks of actions |
| `outer_policy.image_resolution` | [64, 64] | Image resolution |
| `outer_policy.camera_names` | ["global"] | We only use scene cameras for all experiments |
| `env.image_resolution` | [256, 256] | Image resolution of the base environment's camera |
| `sac.batch_size` | 256 | Optimization batch size for the replay buffer |
| `sac.actor_lr` | 0.0001 | Learning rate for the SAC [26] actor network |
| `sac.critic_lr` | 0.0003 | Learning rate for the SAC critic network |
| `sac.temp_lr` | 0.0003 | Learning rate for the SAC entropy coefficient ($\alpha$) |
| `sac.hidden_dims` | [128, 128, 128] | Hidden dimensions for SAC MLP layers |
| `sac.cnn_features` | [32, 32, 32, 32] | Feature maps per layer for the CNN encoder |
| `sac.cnn_strides` | [2, 2, 2, 2] | Strides per layer for the CNN encoder |
| `sac.cnn_padding` | VALID | Padding type for the CNN encoder layers |
| `sac.cnn_latent_dim` | 50 | CNN output feature dim |
| `sac.tau` | 0.005 | Target network soft-update rate |
| `sac.critic_reduction` | min | Ensemble reduction strategy for target Q-values |
| `sac.num_qs` | 10 | Number of Q-networks in the critic ensemble |
| `sac.dropout_rate` | 0.0 | Dropout rate applied inside SAC networks |
| `sac.init_temperature` | 1.0 | Initial value for the SAC entropy coefficient |
| `sac.target_entropy` | -16.0 | Target entropy |
| `sac.action_range` | [-1.0, 1.0] | Defines the allowed min/max action. Note: SAC always sees actions in [-1.0, 1.0], but actions are scaled to be within `sac.action_range` before executing in the environment. |
| `sac.discount` | $1 - \frac{1}{H}$ | Discount is defined based on the task horizon $H$ — see Table I. |
| `train.learning_starts` | *Real:* 10 │ *Sim:* 25 | Episodes collected via uniformly sampling `sac.action_range` to seed the replay buffer pre-learning |
| `train.multi_grad_step` | *Real:* 10 │ *Sim:* 20 | Gradient steps per transition added to the buffer |
| `train.offline_multi_grad_step` | *Real:* 0 │ *Sim:* 5 | Gradient steps per transition added to the buffer for episodes collected before `train.learning_starts` |
| `eval.eval_every` | *Real:* 30 │ *Sim:* 50 | Number of training episodes between evaluations |
| `eval.num_evaluations` | *Real:* 10 │ *Sim:* 64 | Number of evaluation episodes |

TABLE III: **Residual RL Hyperparameters.** Implementations make minimal modifications to Wagenmaker et al. [76] and following standard Residual RL implementations [80, 4, 81]. *Real* indicates experiments run on the WidowX while *Sim* indicates experiments run on Libero-10 [45].

| Hyperparameter | Value | Description |
|---|---|---|
| `outer_policy.action_dim` | *Real:* 7 \| *Sim:* 70 | Action space dimension, which is 7 · `outer_policy.policy_chunk_size`, since Residual RL predicts a correction for each action to be executed |
| `outer_policy.state_dim` | *Real:* 15 \| *Sim:* 78 | Policy has access to an 8-dim proprioceptive observation along with knowledge of the base policy's action |
| `outer_policy.policy_chunk_size` | *Real:* 1 \| *Sim:* 10 | Number of actions used from each evaluation of $\pi$ series policies [63] which generate chunks of actions |
| `outer_policy.image_resolution` | [64, 64] | Image resolution |
| `outer_policy.camera_names` | ["global"] | We only use scene cameras for all experiments |
| `env.image_resolution` | [256, 256] | Image resolution of the base environment's camera |
| `sac.batch_size` | 256 | Optimization batch size for the replay buffer |
| `sac.actor_lr` | 0.0001 | Learning rate for the SAC [26] actor network |
| `sac.critic_lr` | 0.0003 | Learning rate for the SAC critic network |
| `sac.temp_lr` | 0.0003 | Learning rate for the SAC entropy coefficient ($\alpha$) |
| `sac.hidden_dims` | [128, 128, 128] | Hidden dimensions for SAC MLP layers |
| `sac.cnn_features` | [32, 32, 32, 32] | Feature maps per layer for the CNN encoder |
| `sac.cnn_strides` | [2, 2, 2, 2] | Strides per layer for the CNN encoder |
| `sac.cnn_padding` | `VALID` | Padding type for the CNN encoder layers |
| `sac.cnn_latent_dim` | 50 | CNN output feature dim |
| `sac.tau` | 0.005 | Target network soft-update rate |
| `sac.critic_reduction` | `min` | Ensemble reduction strategy for target Q-values |
| `sac.num_qs` | 10 | Number of Q-networks in the critic ensemble |
| `sac.dropout_rate` | 0.0 | Dropout rate applied inside SAC networks |
| `sac.init_temperature` | 1.0 | Initial value for the SAC entropy coefficient |
| `sac.target_entropy` | $-$action_dim$/2$ | Target entropy |
| `sac.action_range` | *Real:* [-0.003, 0.003] *Sim:* [-0.1, 0.1] | Defines the allowed min/max action. Note: SAC always sees actions in [-1.0, 1.0], but actions are scaled to be within `sac.action_range` before executing in the environment. |
| `sac.discount` | $1 - \frac{1}{H}$ | Discount is defined based on the task horizon $H$ — see Table I. |
| `train.human_demos` | *Real:* 3 \| *Sim:* 0 | Human demos collected to seed replay buffer pre-learning |
| `train.learning_starts` | *Real:* 5 \| *Sim:* 25 | Episodes collected via uniformly sampling `sac. action_range` to seed the replay buffer pre-learning (this number does not include any human demos) |
| `train.multi_grad_step` | *Real:* 10 \| *Sim:* 20 | Gradient steps per transition added to the buffer |
| `train.offline_multi_grad_step` | *Real:* 0 \| *Sim:* 5 | Gradient steps per transition added to the buffer for episodes collected before `train. learning_starts` |
| `eval.eval_every` | *Real:* 30 \| *Sim:* 50 | Number of training episodes between evaluations |
| `eval.num_evaluations` | *Real:* 10 \| *Sim:* 64 | Number of evaluation episodes |

TABLE IV: **SARL Hyperparameters.** *Real* indicates experiments run on the WidowX while *Sim* indicates experiments run on Libero-10 [45].

| Hyperparameter | Value | Description |
|---|---|---|
| `outer_policy.max_candidates` | *Real:* 9 │ *Sim:* 4 | Max size of $\mathcal{A}_{\text{sem}}^t$ from Section IV-B and Algorithm 1. Max number of language prompts the algorithm may choose from at each step |
| `outer_policy.cache_size` | *Real:* See Table V *Sim:* 32 | Max size of $\mathcal{A}_{\text{sem}}$ from Section IV-B and Algorithm 1. Total number of unique language prompts the algorithm may learn over |
| `outer_policy.action_dim` | *Real:* See Table V *Sim:* 32 | One-hot encoding over cached language commands |
| `outer_policy.state_dim` | 8 | Policy has access to a proprioceptive observation |
| `outer_policy.policy_chunk_size` | *Real:* 1 │ *Sim:* 10 | Number of actions used from each evaluation of $\pi$ series policies [63] which generate chunks of actions |
| `outer_policy.image_resolution` | [64, 64] | Image resolution |
| `outer_policy.camera_names` | ["global"] | We only use scene cameras for all experiments |
| `env.image_resolution` | [256, 256] | Image resolution of the base environment's camera |
| `sac.batch_size` | 256 | Optimization batch size for the replay buffer |
| `sac.critic_lr` | 0.0003 | Learning rate for the SAC critic network |
| `sac.hidden_dims` | [128, 128, 128] | Hidden dimensions for SAC MLP layers |
| `sac.cnn_features` | [32, 32, 32, 32] | Feature maps per layer for the CNN encoder |
| `sac.cnn_strides` | [2, 2, 2, 2] | Strides per layer for the CNN encoder |
| `sac.cnn_padding` | `VALID` | Padding type for the CNN encoder layers |
| `sac.cnn_latent_dim` | 50 | CNN output feature dim |
| `sac.tau` | 0.005 | Target network soft-update rate |
| `sac.critic_reduction` | `min` | Ensemble reduction strategy for target Q-values |
| `sac.num_qs` | 10 | Number of Q-networks in the critic ensemble |
| `sac.dropout_rate` | 0.0 | Dropout rate applied inside SAC networks |
| `sac.softmax_temp` | See Table V | Softmax temp, $T$, $\pi_{\text{sem}}^t(\ell \mid s) \propto \exp(Q_{\text{sem}}^t(s, \ell)/T)$ from Algorithm 1. |
| `sac.discount` | $1 - \frac{1}{H}$ | Discount is defined based on the task horizon $H$ — see Table I. |
| `train.human_demos` | *Real:* 3 │ *Sim:* 0 | Human demos collected to seed replay buffer pre-learning |
| `train.learning_starts` | *Real:* 5 │ *Sim:* 25 | Episodes collected via uniformly sampling `sac.action_range` to seed the replay buffer pre-learning (this number does not include any human demos) |
| `train.multi_grad_step` | *Real:* 10 │ *Sim:* 20 | Gradient steps per transition added to the buffer |
| `train.offline_multi_grad_step` | *Real:* 0 │ *Sim:* 5 | Gradient steps per transition added to the buffer for episodes collected before `train.learning_starts` |
| `eval.eval_every` | *Real:* 30 │ *Sim:* 50 | Number of training episodes between evaluations |
| `eval.num_evaluations` | *Real:* 10 │ *Sim:* 64 | Number of evaluation episodes |

Fig. 9: Across tasks, an in-context learning VLM [13] takes semantic actions that are semantically meaningful, but lack grounding in physical behaviors, leading to task failures. By contrast, SARL's learned semantic actions correctly navigate scenarios where choosing the right instruction is critical.

## TABLE V: **SARL Task-Specific Hyperparameters**

| Task | outer_policy.cache_size (also outer_policy.action_dim) | outer_policy.softmax_temp |
|---|---|---|
| **Real Task 1:** Move the hammer to the plate. Then, grasp the mushroom. | 64 | 0.2 |
| **Real Task 2:** Move the banana to the pot on the right and sushi to the bowl. | 100 | 0.5 |
| **Real Task 3:** Move the left stuffed toy to the largest bowl. | 32 | 0.1 |
| **Real Task 4:** Move the pot lid to the towel. | 32 | 0.2 |
| **Libero-10** | 32 | 0.03 |

You are guiding a robot to complete the task '[self.task_instruction]' in its environment by generating candidate subtask instructions for the robot to follow given an image of the robot's current observation. There are several kinds of instructions the robot understands, each at a different level of abstraction, detailed below. It is not known apriori which level of abstraction will induce the desired behavior in the robot's low-level language conditioned policy, so your job is to generate candidates that will be processed by a downstream algorithm.
Below are examples of the kinds of instructions you can generate for the robot.
Some examples of High-level instructions are:
– 'grasp the mushroom in the pot'
– 'wipe the cloth across the table'
– 'move the lid above the pot'
– 'lift the carrot up' # Lifting is generallly good post-grasping
– 'move towards the stuffed toy'
– 'move the eggplant to the bowl'
The robot is generally reliable at following high-level commands, especially when interacting with non-novel objects. They can also be used to better understand the scene and robot state, such as by saying 'lift the object' to ascertain if the robot has actually firmly grasped it. However, these commands may fail when interacting with novel objects, as the robot may not recognize them by name. The robot also may also confuse similar objects with each other (e.g., it may mix up a bowl and pot, or multiple instances of the same kind of object).
Some examples of Atomic Motion instructions are:
– 'move left'
– 'move above the bowl'
– 'move down and close'
– 'open the gripper'
– 'move the cloth left'
– 'pull the drawer handle backward'
Motion commands are good for general positioning, such as moving to a pre-grasp pose above the object the robot must interact with. They can also be used for positional corrections, e.g., 'move up and right above the bowl' if the gripper is too low. These commands are also appropriate when the task involves spatial reasoning (e.g., tasks that reference left or right). However, these commands are often underspecified, so the robot may reach for the wrong object if simply told 'move to the right and grasp.'
Note that to go to the back of the table, tell the robot to go 'forwards' (it is forwards from the robot's perspective).
Finally, some examples of Combination instructions are:
– 'move down to the mushroom and grasp it'
– 'grab the mushroom and move it to the plate on the left'
– 'open the gripper and move right'
Combination commands are useful for supplementing the weaknesses of other command types, such as saying 'move right to grasp the carrot' rather than just 'move right and grasp' (if there are multiple objects to the right of the robot) or 'grasp the carrot on the right' (if there are multiple carrots in the scene). However, being too specific may result in out-of-distribution commands, which the robot policy fails to follow. Additionally, less specific commands can be useful for positioning the robot in a location where it's more likely to succeed.
You MUST follow these steps when generating your response:
– Step 1: Briefly (1-2 sentences) describe the scene in the current observation, such as enumerating which objects are present and the robot grippers current state. If the robot's gripper is close to an object, think carefully about whether the object has already been grasped, lifted or neither. Think about whether the gripper is currently open or closed.
– Step 2: Briefly (1-2 sentences) enumerate the robot's full sequence of objectives required to complete the task.
– Step 3: Briefly (1 sentences) turn the robot's immediate next objective into a subtask statement. A relatively longer-horizon subtask statement is preferred in this step. Eg. "pick up the egg and move it to the bowl" rather than "grasp the egg" or "move down" or "close the gripper". The subtask objective should make sense to execute immediately given the current state.
– Step 4: Briefly (1-2 sentences) think about 1) the overall direction the robot should move in (left, right, back of table ["forwards"], toward the camera ["backwards"], up, down) and 2) obvious verbs that could be used to instruct the robot (use verbs likely present in robot datasets, without being too extravagant and be sure to PRIORITIZE verbs found in the original task description), whether the robot should open/close the gripper.
– Step 5: (1-2 sentences) think about how to avoid these common failure modes: confusing objects with the same name, interacting with the object directly in front of the robot rather than the intended object, and executing parts of the command out of order (eg. dropping object prematurely when the command involves both "moving" and "releasing").
– Step 6: Output the FINAL MASTER LIST of commands for the robot to choose between. The master list should follow this guidance:
– Output commands most likely to help the robot succeed, while still being diverse.
– Output exactly eight commands. Always output EXACTLY TWO SIMPLE atomic motion commands ouptut more combination commands than any other abstraction type.
– Commands should be diverse in horizon: shorter horizon (eg. "move above the egg") and longer-horizon behaviors ("move the egg to the pot"), but should not exceed the length of the next subtask.
– Use an indicator (left, right, blue) to distinguish objects of the same name ONLY if the robot is near the wrong copy of that object. DO NOT use the indicator if the robot is currently interacting with the correct copy.
– All commands will be executed immediately, so do not output forward-looking commands that do not make sense to run at the robot's current state (eg. "open the gripper" does not make sense if the robot is not yet at the correct position to release the object). However, "move the egg to the plate" does make sense if consistent with the task).
The master list should be numbered. DO NOT organize the master list by objective, simply list all the commands together in one numbered list. After the master list, DO NOT OUTPUT ANY OTHER TEXT.
Current Obervation: [current_observation]

Fig. 10: Prompt for open-ended real-world semantic action candidate generation

You are guiding a robot to complete the task '[self.task_instruction]' in its environment by chosing a list of candidate task instructions for the robot to follow given an image of the robot's current observation. There are several kinds of instructions the robot understands, each at a different level of abstraction, detailed below. It is not known apriori which level of abstraction will induce the desired behavior in the robot's low-level language conditioned policy, so your job is to generate candidates that will be processed by a downstream algorithm.
Below are examples of the kinds of instructions you can generate for the robot.
Some examples of High-level instructions are:
- 'grasp the mushroom in the pot'
- 'wipe the cloth across the table'
- 'move the lid above the pot'
- 'lift the carrot up'  Lifting is generallly good post-grasping
- 'move towards the stuffed toy'
- 'move the eggplant to the bowl'
The robot is generally reliable at following high-level commands, especially when interacting with non-novel objects. They can also be used to better understand the scene and robot state, such as by saying 'lift the object' to ascertain if the robot has actually firmly grasped it. However, these commands may fail when interacting with novel objects, as the robot may not recognize them by name. The robot also may also confuse similar objects with each other (e.g., it may mix up a bowl and pot, or multiple instances of the same kind of object).
Some examples of Atomic Motion instructions are:
- 'move left'
- 'move above the bowl'
- 'move down and close'
- 'open the gripper'
- 'move the cloth left'
- 'pull the drawer handle backward'
Motion commands are good for general positioning, such as moving to a pre-grasp pose above the object the robot must interact with. They can also be used for positional corrections, e.g., 'move up and right above the bowl' if the gripper is too low. These commands are also appropriate when the task involves spatial reasoning (e.g., tasks that reference left or right). However, these commands are often underspecified, so the robot may reach for the wrong object if simply told 'move to the right and grasp.' Note that to go to the back of the table, tell the robot to go 'forwards' (it is forwards from the robot's perspective).
Finally, some examples of Combination instructions are:
- 'move down to the mushroom and grasp it'
- 'grab the mushroom and move it to the plate on the left'
- 'open the gripper and move right'
Combination commands are useful for supplementing the weaknesses of other command types, such as saying 'move right to grasp the carrot' rather than just 'move right and grasp' (if there are multiple objects to the right of the robot) or 'grasp the carrot on the right' (if there are multiple carrots in the scene). However, being too specific may result in out-of-distribution commands, which the robot policy fails to follow. Additionally, less specific commands can be useful for positioning the robot in a location where it's more likely to succeed.
You MUST follow these steps when generating your response:
- Step 1: Briefly (1-2 sentences) describe the scene in the current observation, such as enumerating which objects are present and the robot grippers current state. If the robot's gripper is close to an object, think carefully about whether the object has already been grasped, lifted or neither. Think about whether the gripper is currently open or closed.
- Step 2: Briefly (1-2 sentences) enumerate the robot's full sequence of objectives required to complete the task. IMPORTANT: ONCE A SUBGOAL HAS BEEN COMPLETED, DO NOT KEEP INTERACTING WITH THE OBJECT FOR THAT SUBGOAL. THE DEFINITION OF TASK SUCCESS IS QUITE LOOSE (eg. if the goal is to put an eggplant on the towel, ANY part of the eggplant being on the towel will be considered a success. DO NOT keep interacting with the eggplant).
- Step 3: Briefly (1 sentences) turn the robot's immediate next objective into a subtask statement. A relatively longer-horizon subtask statement is preferred in this step. Eg. "pick up the egg and move it to the bowl" rather than "grasp the egg" or "move down" or "close the gripper". The subtask objective should make sense to execute immediately given the current state.
- Step 4: Briefly (1-2 sentences) think about 1) the overall direction the robot should move in (left, right, back of table ["forwards"], toward the camera ["backwards"], up, down) and 2) obvious verbs that could be used to instruct the robot (use verbs likely present in robot datasets, without being too extravagant and be sure to PRIORITIZE verbs found in the original task description), whether the robot should open/close the gripper.
- Step 5: (1-2 sentences) think about how to avoid these common failure modes: confusing objects with the same name, interacting with the object directly in front of the robot rather than the intended object, and executing parts of the command out of order (eg. dropping object prematurely when the command involves both "moving" and "releasing").
- Step 6: Output the FINAL MASTER LIST of commands for the robot to choose between. The master list should follow this guidance:
- Output exactly eight commands. Commands MUST BE EXACTLY THE SAME AS THOSE ON THE LIST PROVIDED BELOW. DO NOT OUTPUT ANY OTHER COMMANDS. The list may not be perfect, choose the best fit.
- Output commands most likley to help the robot succeed, while still being diverse.
- Always output EXACTLY TWO SIMPLE atomic motion commands ouptut more combination commands than any other abstraction type. YOU MUST ALWAYS INCLUDE EITHER "move right" or "move left" in the list.
- Commands should be diverse in horizon: shorter horizon (eg. "move above the egg") and longer-horizon behaviors ("move the egg to the pot"), but should not exceed the length of the next subtask.
- Use an indicator (left, right, blue) to distinguish objects of the same name ONLY if the robot is near the wrong copy of that object. DO NOT use the indicator if the robot is currently interacting with the correct copy.
The master list should be numbered (1. 2. 3. ...) but use only the text of the command exactly as presented. DO NOT organize the master list by objective, simply list all the commands together in one numbered list. After the master list, DO NOT OUTPUT ANY OTHER TEXT.
Current Obervation: [current_observation]
You must choose commands EXACTLY from this list: [cached_commands_str]

Fig. 11: Prompt for real-world semantic action candidate generation from cache

You are guiding a robot to complete a task by generating a subtask instruction for the robot to follow given an image of the robot's current observation, and a history of observations and past commands. The robot is trying to accomplish the overall task: '[self.task_description]'. You should keep the rest of the environment unchanged. Note that all objects required for the task are present (no need to search for objects). There are several kinds of instructions the robot understands, each at a different level of abstraction, detailed below. Sometimes, a high-level instruction is sufficient for the robot to make progress towards solving the task, but other times, a more specific command is required.
Below are examples of the kinds of instructions you can generate for the robot.
Some examples of High-level instructions are:
– 'grasp the mushroom in the pot'
– 'wipe the cloth across the table'
– 'move the lid above the pot'
– 'lift the carrot up'
– 'move towards the stuffed toy'
– 'drop the eggplant in the bowl'
The robot is generally reliable at following high-level commands, especially when interacting with non-novel objects. They can also be used to better understand the scene and robot state, such as by saying 'lift the object' to ascertain if the robot has actually firmly grasped it. However, these commands may fail when interacting with novel objects, as the robot may not recognize them by name. The robot also may also confuse similar objects with each other (e.g., it may mix up a bowl and pot, or multiple instances of the same kind of object).
Some examples of Atomic Motion instructions are:
– 'move left'
– 'move right and down'
– 'move down and close'
– 'open the gripper'
– 'move the cloth left'
– 'pull the drawer handle backward'
Motion commands are good for general positioning, such as moving to a pre-grasp pose above the object the robot must interact with. They can also be used for more minor corrections, e.g., 'move up and right above the bowl' if the gripper is too low. These commands are also appropriate when the task involves spatial reasoning (e.g., tasks that reference left or right) or moving in unconventional ways (e.g., lifting the arm to grasp something on a high shelf). However, these commands are often underspecified, so the robot may reach for the wrong object if simply told 'move to the right and grasp.'
Some examples of Position-based instructions are:
– 'pick up the object at <mushroom>'
– 'go to <pot>'
– 'grasp at <blue cube>'
– 'move above <plate>'
– 'open gripper at <bowl>'
– 'move towards <towel>'
Note, you simply need to specify the object name within angle brackets, e.g., <mushroom>, and a object detector system will fill in the pixel coordinates for you. If there are multiple instances of the object, provide a unique, specific identifier within the brackets to indicate which instance you are referring to, e.g., <mushroom on the left>.
Position-based commands are especially helpful for specifying novel objects that the robot fails to recognize by name. They can also be used to clearly specify locations, such as where to place or move objects. However, these commands can be unreliable, as the robot may get confused (e.g., reaching for the wrong object in scenes with clutter or with multiple instances of the same object). The object detector may also fail, especially when objects are occluded.
Next, some examples of Combination instructions are:
– 'move down to the object at <mushroom> and grasp it'
– 'grab the mushroom at <mushroom on the left> and move it to <plate>'
– 'move the stuffed toy toward the box at <box>'
Combination commands are useful for supplementing the weaknesses of other command types, such as saying 'move right to grasp the object at <object description>' rather than just 'move right and grasp' (if there are multiple objects to the right of the robot). However, being too specific may result in out-of-distribution commands, which the robot policy fails to follow. Additionally, less specific commands can be useful for positioning the robot in a location where it's more likely to succeed.
You MUST follow these steps when generating your response:
– Step 1: Briefly (1-2 sentences) describe the scene in the current observation, and any important differences from past observations (eg. objects moved, picked up, dropped etc).
– Step 2: Briefly (1 sentence) describe what must happen next to make progress towards the overall task.
– Step 3: Briefly reason (2-3 sentences) about which level of abstraction is most appropriate given the current observation and task progress.
– Step 4: Determine the final steerable command (last line) for the robot to execute. Only the LAST line of your output will be interpreted as the subtask command. In the LAST line, answer ONLY with the subtask instruction that the robot should execute, and NOTHING ELSE.
Note: The robot will ask you for a new instruction after executing for a few steps. If you've issued the same type of command more than once without progress, you MUST issue a command at a different level of abstraction. If you previously provided a command with positions <> (without progress), you MAY NOT use positions in the next command. For example:
– If you previously tried to pick up the mushroom by specifying the mushroom's position, but the robot did not reach the mushroom, next time you can explicitly tell the robot to move in a certain direction (eg. move to the right and down).
– If the robot reached for the wrong object when you told it to pick up the mushroom at <mushroom on the left>. This time, you should give a command at a different level of abstraction: move left to the mushroom and grasp it.
Here are the past observations and commands: [history]
Current Obervation: [current_observation]
Based on this observation, generate the next command. Remember the instructions: describe the scene, determine what must happen next, determine the best level of abstraction, output the final command. Again, the LAST line of your output will be interpreted as the final steerable, subtask command. In the LAST line, answer ONLY with the subtask instruction that the robot should execute, and NOTHING ELSE.

Fig. 12: Prompt for real-world in-context learning VLM baseline

You are guiding a robot to complete the task '[self.task_instruction]' in its environment by generating candidate subtask instructions for the robot to follow given an image of the robot's current observation. There are several kinds of instructions the robot understands, each at a different level of abstraction, detailed below. It is not known apriori which level of abstraction will induce the desired behavior in the robot's low-level language conditioned policy, so your job is to generate candidates that will be processed by a downstream algorithm.
Below are examples of the kinds of instructions you can generate for the robot.
Some examples of High-level instructions are:
- 'grasp the mushroom in the pot'
- 'wipe the cloth across the table'
- 'move the lid above the pot'
- 'lift the carrot up'  Lifting is generallly good post-grasping
- 'move towards the stuffed toy'
- 'move the eggplant to the bowl'
The robot is generally reliable at following high-level commands, especially when interacting with non-novel objects. They can also be used to better understand the scene and robot state, such as by saying 'lift the object' to ascertain if the robot has actually firmly grasped it. However, these commands may fail when interacting with novel objects, as the robot may not recognize them by name. The robot also may also confuse similar objects with each other (e.g., it may mix up a bowl and pot, or multiple instances of the same kind of object).
Some examples of Atomic Motion instructions are:
- 'move left'
- 'move above the bowl'
- 'move down and close'
- 'open the gripper'
- 'move the cloth left'
- 'pull the drawer handle backward'
Motion commands are good for general positioning, such as moving to a pre-grasp pose above the object the robot must interact with. They can also be used for positional corrections, e.g., 'move up and right above the bowl' if the gripper is too low. These commands are also appropriate when the task involves spatial reasoning (e.g., tasks that reference left or right). However, these commands are often underspecified, so the robot may reach for the wrong object if simply told 'move to the right and grasp.'
Note that to go to the back of the table, tell the robot to go 'forwards' (it is forwards from the robot's perspective).
Next, some examples of Combination instructions are:
- 'move down to the mushroom and grasp it'
- 'grab the mushroom and move it to the plate on the left'
- 'open the gripper and move right'
Combination commands are useful for supplementing the weaknesses of other command types, such as saying 'move right to grasp the carrot' rather than just 'move right and grasp' (if there are multiple objects to the right of the robot) or 'grasp the carrot on the right' (if there are multiple carrots in the scene). However, being too specific may result in out-of-distribution commands, which the robot policy fails to follow. Additionally, less specific commands can be useful for positioning the robot in a location where it's more likely to succeed.
Finally, there is the 'reset to home' command, which can be issued when the robot is stuck. It is useful for repositioning the robot to a known state.
You MUST follow these steps when generating your response:
- Step 1: Briefly (1-2 sentences) describe the scene in the current observation, such as enumerating which objects are present and the robot grippers current state. If the robot's gripper is close to an object, think carefully about whether the object has already been grasped, lifted or neither. Think about whether the gripper is currently open or closed.
- Step 2: Briefly (1-2 sentences) enumerate the robot's full sequence of objectives required to complete the task.
- Step 3: Briefly (1 sentences) turn the robot's immediate next objective into a subtask statement. A relatively longer-horizon subtask statement is preferred in this step. Eg. "pick up the egg and move it to the bowl" rather than "grasp the egg" or "move down" or "close the gripper". The subtask objective should make sense to execute immediately given the current state.
- Step 4: Briefly (1-2 sentences) think about 1) the overall direction the robot should move in (left, right, forwards, backwards, up, down) and 2) obvious verbs that could be used to instruct the robot (use verbs likely present in robot datasets, without being too extravagant and be sure to PRIORITIZE verbs found in the original task description).
- Step 5: (1-2 sentences) think about how to avoid these common failure modes: confusing objects with the same name, interacting with the object directly in front of the robot rather than the intended object, and executing parts of the command out of order (eg. dropping object prematurely when the command involves both "moving" and "releasing").
- Step 6: Output the FINAL MASTER LIST of commands for the robot to choose between. The master list should follow this guidance:
- Output commands most likley to help the robot succeed, while still being diverse.
- Output exactly four commands. Always output a single atomic motion commands ouptut more subtask commands than any other abstraction type.
- Commands should be diverse in horizon: shorter horizon (eg. "move above the egg") and longer-horizon behaviors ("move the egg to the pot"), but should not exceed the length of the next subtask.
- Use an indicator (left, right, blue) to distinguish objects of the same name ONLY if the robot is near the wrong copy of that object. DO NOT use the indicator if the robot is currently interacting with the correct copy.
- All commands will be executed immediately, so do not output forward-looking commands that do not make sense to run at the robot's current state (eg. "open the gripper" does not make sense if the robot is not yet at the correct position to release the object. However, "move the egg to the plate" does make sense if consistent with the task).
The master list should be numbered. DO NOT organize the master list by objective, simply list all the commands together in one numbered list. After the master list, DO NOT OUTPUT ANY OTHER TEXT.
Current Obervation: [current_observation]

Fig. 13: Prompt for open-ended sim semantic action candidate generation

You are guiding a robot to complete the task '[self.task_instruction]' in its environment by chosing a list of candidate task instructions for the robot to follow given an image of the robot's current observation. There are several kinds of instructions the robot understands, each at a different level of abstraction, detailed below. It is not known apriori which level of abstraction will induce the desired behavior in the robot's low-level language conditioned policy, so your job is to generate candidates that will be processed by a downstream algorithm.
Below are examples of the kinds of instructions you can generate for the robot.
Some examples of High-level instructions are:
- 'grasp the mushroom in the pot'
- 'wipe the cloth across the table'
- 'move the lid above the pot'
- 'lift the carrot up'  Lifting is generallly good post-grasping
- 'move towards the stuffed toy'
- 'move the eggplant to the bowl'
The robot is generally reliable at following high-level commands, especially when interacting with non-novel objects. They can also be used to better understand the scene and robot state, such as by saying 'lift the object' to ascertain if the robot has actually firmly grasped it. However, these commands may fail when interacting with novel objects, as the robot may not recognize them by name. The robot also may also confuse similar objects with each other (e.g., it may mix up a bowl and pot, or multiple instances of the same kind of object).
Some examples of Atomic Motion instructions are:
- 'move left'
- 'move above the bowl'
- 'move down and close'
- 'open the gripper'
- 'move the cloth left'
- 'pull the drawer handle backward'
Motion commands are good for general positioning, such as moving to a pre-grasp pose above the object the robot must interact with. They can also be used for positional corrections, e.g., 'move up and right above the bowl' if the gripper is too low. These commands are also appropriate when the task involves spatial reasoning (e.g., tasks that reference left or right). However, these commands are often underspecified, so the robot may reach for the wrong object if simply told 'move to the right and grasp.' Note that to go to the back of the table, tell the robot to go 'forwards' (it is forwards from the robot's perspective).
Next, some examples of Combination instructions are:
- 'move down to the mushroom and grasp it'
- 'grab the mushroom and move it to the plate on the left'
- 'open the gripper and move right'
Combination commands are useful for supplementing the weaknesses of other command types, such as saying 'move right to grasp the carrot' rather than just 'move right and grasp' (if there are multiple objects to the right of the robot) or 'grasp the carrot on the right' (if there are multiple carrots in the scene). However, being too specific may result in out-of-distribution commands, which the robot policy fails to follow. Additionally, less specific commands can be useful for positioning the robot in a location where it's more likely to succeed.
Finally, there is the 'reset to home' command, which can be issued when the robot is stuck. It is useful for repositioning the robot to a known state.
You MUST follow these steps when generating your response:
- Step 1: Briefly (1-2 sentences) describe the scene in the current observation, such as enumerating which objects are present and the robot grippers current state. If the robot's gripper is close to an object, think carefully about whether the object has already been grasped, lifted or neither. Think about whether the gripper is currently open or closed.
- Step 2: Briefly (1-2 sentences) enumerate the robot's full sequence of objectives required to complete the task.
- Step 3: Briefly (1 sentences) turn the robot's immediate next objective into a subtask statement. A relatively longer-horizon subtask statement is preferred in this step. Eg. "pick up the egg and move it to the bowl" rather than "grasp the egg" or "move down" or "close the gripper". The subtask objective should make sense to execute immediately given the current state.
- Step 4: Briefly (1-2 sentences) think about 1) the overall direction the robot should move in (left, right, forwards, backwards, up, down) and 2) obvious verbs that could be used to instruct the robot (use verbs likely present in robot datasets, without being too extravagant and be sure to PRIORITIZE verbs found in the original task description).
- Step 5: (1-2 sentences) think about how to avoid these common failure modes: confusing objects with the same name, interacting with the object directly in front of the robot rather than the intended object, and executing parts of the command out of order (eg. dropping object prematurely when the command involves both "moving" and "releasing").
- Step 6: Output the FINAL MASTER LIST of commands for the robot to choose between. The master list should follow this guidance:
- Output exactly four commands. Commands MUST BE EXACTLY THE SAME AS THOSE ON THE LIST PROVIDED BELOW. DO NOT OUTPUT ANY OTHER COMMANDS. The list may not be perfect, choose the best fit.
- Output commands most likley to help the robot succeed, while still being diverse.
- Always output a single atomic motion commands ouptut more subtask commands than any other abstraction type.
- Commands should be diverse in horizon: shorter horizon (eg. "move above the egg") and longer-horizon behaviors ("move the egg to the pot"), but should not exceed the length of the next subtask.
- Use an indicator (left, right, blue) to distinguish objects of the same name ONLY if the robot is near the wrong copy of that object. DO NOT use the indicator if the robot is currently interacting with the correct copy.
- All commands will be executed IMMEDIATELY (THIS IS VERY IMPORTANT), so do not output forward-looking commands that do not make sense to run at the robot's current state (eg. "open the gripper" does not make sense if the robot is not yet at the correct position to release the object. However, "move the egg to the plate" does make sense if consistent with the task). The master list should be numbered. DO NOT organize the master list by objective, simply list all the commands together in one numbered list. After the master list, DO NOT OUTPUT ANY OTHER TEXT.
Current Obervation: [current_observation]
You must choose commands EXACTLY from this list: [cached_commands_str]

Fig. 14: Prompt for sim semantic action candidate generation from cache

You are guiding a robot to complete a task by generating a subtask instruction for the robot to follow given an image of the robot's current observation, and a history of observations and past commands. The robot is trying to accomplish the overall task: '[self.task_description]'. You should keep the rest of the environment unchanged. Note that all objects required for the task are present (no need to search for objects). There are several kinds of instructions the robot understands, each at a different level of abstraction, detailed below. Sometimes, a high-level instruction is sufficient for the robot to make progress towards solving the task, but other times, a more specific command is required.
Below are examples of the kinds of instructions you can generate for the robot.
Some examples of High-level instructions are:
- 'grasp the mushroom in the pot'
- 'wipe the cloth across the table'
- 'move the lid above the pot'
- 'lift the carrot up'  IMPORTANT: Lifting is generallly good post-grasping
- 'move towards the stuffed toy'
- 'drop the eggplant in the bowl'
The robot is generally reliable at following high-level commands, especially when interacting with non-novel objects. They can also be used to better understand the scene and robot state, such as by saying 'lift the object' to ascertain if the robot has actually firmly grasped it. However, these commands may fail when interacting with novel objects, as the robot may not recognize them by name. The robot also may also confuse similar objects with each other (e.g., it may mix up a bowl and pot, or multiple instances of the same kind of object).
Some examples of Atomic Motion instructions are:
- 'move left'
- 'move right and down'
- 'move down and close'
- 'open the gripper'
- 'move the cloth left'
- 'pull the drawer handle backward'
Motion commands are good for general positioning, such as moving to a pre-grasp pose above the object the robot must interact with. They can also be used for more minor corrections, e.g., 'move up and right above the bowl' if the gripper is too low. These commands are also appropriate when the task involves spatial reasoning (e.g., tasks that reference left or right) or moving in unconventional ways (e.g., lifting the arm to grasp something on a high shelf). However, these commands are often underspecified, so the robot may reach for the wrong object if simply told 'move to the right and grasp.'
Next, some examples of Combination instructions are:
- 'move down to the object and grasp it'
- 'grab the mushroom and move it to the plate'
- 'move the stuffed toy toward the box and drop it in'
Combination commands are useful for supplementing the weaknesses of other command types, such as saying 'move right to grasp the object' rather than just 'grasp the object' (if there are multiple objects near the robot). However, being too specific may result in out-of-distribution commands, which the robot policy fails to follow. Additionally, less specific commands can be useful for positioning the robot in a location where it's more likely to succeed.
Finally, there is the 'reset to home' command, which can be issued when the robot is stuck. It is useful for repositioning the robot to a known state. Sometimes, this command needs to be issues multiple times in a row to get the robot unstuck.
You MUST follow these steps when generating your response:
- Step 1: Briefly (1-2 sentences) describe the scene in the current observation, and any important differences from past observations (eg. objects moved, picked up, dropped etc).
- Step 2: Briefly (1 sentence) describe what must happen next to make progress towards the overall task.
- Step 3: Briefly reason (2-3 sentences) about which level of abstraction is most appropriate given the current observation and task progress.
- Step 4: Determine the final steerable command (last line) for the robot to execute. Only the LAST line of your output will be interpreted as the subtask command. In the LAST line, answer ONLY with the subtask instruction that the robot should execute, and NOTHING ELSE.
Note: The robot will ask you for a new instruction after executing for a few steps. If you've issued the same type of command more than once without progress, you MUST issue a command at a different level of abstraction. For example:
- If you previously tried to pick up the mushroom by specifying the mushroom by name, but the robot did not reach the mushroom, next time, for example, you can explicitly tell the robot to move in a certain direction (eg. move to the right and down).
Here are the past observations and commands: [history]
Current Obervation: [current_observation]
Based on this observation, generate the next command. Remember the instructions: describe the scene, determine what must happen next, determine the best level of abstraction, output the final command. Again, the LAST line of your output will be interpreted as the final steerable, subtask command. In the LAST line, answer ONLY with the subtask instruction that the robot should execute, and NOTHING ELSE.

Fig. 15: Prompt for sim in-context learning VLM baseline