# OpenReview forum: "Adapting Generalist Robot Policies with Semantic Reinforcement Learning"
_roboticsfoundation.org/RSS/2026/Workshop/RL4VLA — RL4VLA_

### Official Review · Reviewer_CfoE · 2026-06-26
**Promising idea, but brittle in practice**

**Rating:** 6
**Confidence:** 5

**Review:**

# Summary

This paper proposes SARL, a method that adapts VLA policies by running reinforcement learning over language prompts rather than low-level robot actions. A VLM proposes candidate semantic actions, and SARL learns which prompts can steer the frozen VLA toward task success.

# **Strengths and weaknesses**

## **Strengths**:

* The paper is highly relevant to the workshop. It directly studies how to improve generalist VLA policies through online RL, which is an important problem for deployment-time robot learning.

*
  Treating language prompts as semantic actions is a compelling direction. It avoids directly exploring low-level robot actions and instead leverages skills already encoded in the VLA. This makes the method conceptually suitable for long-horizon tasks that require composing multiple short skills.

*
  The real-world WidowX experiments are valuable. The results suggest that prompt-level adaptation can sometimes unlock behaviors that low-level residual or latent-space steering methods struggle to find.

*
  The paper clearly shows that VLM-generated commands may be semantically reasonable but physically ungrounded for a specific VLA. This supports the motivation for learning from real interaction rather than relying only on VLM decomposition.

## **Weaknesses**:

### **Brittle action space construction**

The main weakness is the construction of the semantic action space. Although the paper describes SARL as RL over language prompts, the actual implementation relies on a VLM to generate candidate prompts, a cache of the first \(k\) prompts, one-hot prompt representations, task-specific cache sizes, and tuned softmax temperatures. These choices introduce many hidden variables. If the early cache does not contain the right commands, SARL may not be able to discover them later. This makes the method appear brittle and highly dependent on prompt/action-space engineering.

### **Human demonstrations are a strong assumption**

The real-world experiments use human demonstrations to seed the replay buffer. This is not a minor detail. Since the paper itself shows that reasonable prompts can fail due to poor grounding, humans may need substantial trial and error to discover effective command sequences. The paper does not compare SARL with and without these demonstrations, so it is unclear how much of the sample efficiency comes from RL versus manually discovered semantic trajectories.

### **Unclear practical advantage over high-level imitation learning**

Given that humans already provide semantic demonstrations, a natural baseline would be behavior cloning or imitation learning over the same high-level prompt space. The paper does not include this comparison. Therefore, it is unclear whether online RL provides significant benefit beyond training a high-level prompt selector from human-provided semantic demos (with the same effort budget of tuning all the things for one single RL training).

### **High cost**

SARL requires repeated VLM calls during training, which can substantially increase wall-clock time and deployment cost. The appendix introduces caching and candidate reuse to reduce this burden, but these are additional engineering tricks rather than a fundamental solution. The paper reports episode-level or environment-step efficiency, but does not clearly report total VLM calls, latency, wall-clock training time, or API/computation cost.

### **Strong dependence on the base VLA**

The method only works if the underlying VLA already contains a diverse set of useful prompt-inducible skills. SARL does not learn genuinely new low-level capabilities; it mainly learns how to select among behaviors the VLA can already express. This limits the significance of the method in settings where the VLA is weaker or where the target task requires skills outside the pretrained repertoire.

### **Insufficient ablations**

Many crucial components are not sufficiently ablated, including VLM choice, VLM prompt design, cache size, candidate set size, query frequency, human demonstrations, and task-specific hyperparameters. Without these studies, it is difficult to know whether the reported gains come from a robust semantic RL mechanism or from careful engineering around the action space.

# **Overall assessment**

The paper is original and relevant, and the high-level idea is promising. However, the current implementation feels more like a carefully engineered proof of concept than a practical general-purpose VLA adaptation method. Its success appears to require a strong VLA, good VLM-generated candidates, useful human demonstrations, and significant prompt/cache tuning. As a result, the method has high potential under favorable conditions, but its robustness, reproducibility, and practical deployability remain insufficiently demonstrated.

---

### Official Review · Reviewer_MZ7w · 2026-06-28
**good paper with good idea of optimizing prompt space for RL-finutining of VLAs**

**Rating:** 6
**Confidence:** 4

**Review:**

This paper proposes a learning framework for steering a frozen VLA model to solve tasks by optimizing its prompt space through real-world interactions; its topic fits this workshop. It is also well-written, and its comparison with residual RL, DSRL, base VLA prompting, and ICL-VLM prompting supports the claim that online grounding of prompts can outperform purely semantic task decomposition.

This framework strongly depends on a VLM proposing good-quality candidate prompts, however their diversity and possible failure modes are not deeply analysed here and it seems the assumption that vla contains the needed primitive skills is also a big obstacle to scaling this approach up, e.g., to open-ended prompt spaces or across more robots.

---

### Decision · Program_Chairs · 2026-07-03

**Decision:**

Accept

**Comment:**

This paper introduces SARL, a reinforcement learning framework that adapts frozen generalist robot policies by optimizing language prompts instead of actions, enabling efficient learning of complex long-horizon tasks through semantic skill composition. The reviewers raised valid concerns regarding some limitations of the proposed approach. While we do not consider these issues sufficient to reject the paper, we encourage the authors to discuss them in the camera-ready version.